# TWO-TAILED AVERAGING: ANYTIME ADAPTIVE ONCE-IN-A-WHILE OPTIMAL WEIGHT AVERAGING FOR BETTER GENERALIZATION

## ABSTRACT

Tail averaging improves on Polyak averaging's non-asymptotic behaviour by excluding a number of leading iterates of stochastic optimization from its calculations. In practice, with a finite number of optimization steps and a learning rate that cannot be annealed to zero, tail averaging can get much closer to a local minimum point of the training loss than either the individual iterates or the Polyak average. However, the number of leading iterates to ignore is an important hyperparameter, and starting averaging too early or too late leads to inefficient use of resources or suboptimal solutions. Our work focusses on improving generalization, which makes setting this hyperparameter even more difficult, especially in the presence of other hyperparameters and overfitting. Furthermore, before averaging starts, the loss is only weakly informative of the final performance, which makes early stopping unreliable. To alleviate these problems, we propose an anytime variant of tail averaging intended for improving generalization not pure optimization, that has no hyperparameters and approximates the optimal tail at all optimization steps. Our algorithm is based on two running averages with adaptive lengths bounded in terms of the optimal tail length, one of which achieves approximate optimality with some regularity. Requiring only the additional storage for two sets of weights and periodic evaluation of the loss, the proposed two-tailed averaging algorithm is a practical and widely applicable method for improving generalization.

## 1 INTRODUCTION

For the series of iterates produced by Stochastic Gradient Descent (SGD) (Robbins and Monro, 1985) to converge to a local minimum point of the training loss, the learning rate must be annealed to zero. Polyak averaging (Polyak and Juditsky, 1992; Ruppert, 1988) improves on SGD and achieves a statistically optimal convergence rate by averaging all iterates to produce the final solution. Tail or suffix averaging (Jain et al., 2018; Rakhlin et al., 2011) takes this further and improves the non-asymptotic behaviour by dropping a number of leading iterates from the average, speeding up the decay of the effect of the initial state while allowing the learning rate to stay constant. Both of these properties are advantageous in practice, where a finite number of optimization steps are taken, and because large learning rates may bias optimization towards flatter and wider minima, which improves generalization (Hochreiter and Schmidhuber, 1997; Keskar et al., 2016). Focussing on large learning rates, flat minima, and generalization, Izmailov et al. (2018) propose Stochastic Weight Averaging (SWA), which takes the same form as tail averaging but is motivated from an ensembling point of view.

Tail averaging starts after a given number of optimization steps. Setting this hyperparameter to minimize the training loss already poses some difficulties, which only become more pronounced and numerous in the context of generalization, our primary focus in this work.

- Triggering averaging too early is inefficient as the average must grow long for the early weights to matter less.
- Triggering averaging too late is inefficient as it does not use valuable information.
- Due to interdependencies, tuning of other hyperparameters may become harder.

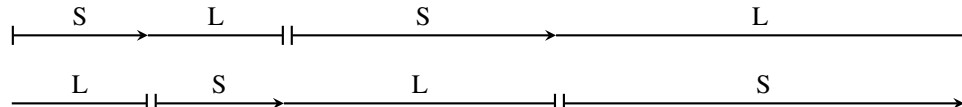

Figure 1: Example evolution of the two running averages of weights over optimization steps. Continuous lines indicate optimization iterates being added to averages over the course of optimization. There is always one short (S) and one long (L) average, with the long one having more iterates averaged and a better loss. When the loss with the short one would become better, the short average is renamed to long (marked by arrowheads), and the short one is restarted from an empty state (marked by discontinuities between double vertical lines). The long average thus always starts from the final state of the preceeding short average and incorporates more iterates until the new short average becomes at least as good in terms of the loss. In any interval labeled with L (from an arrowhead to a vertical bar), there is at least one point where the length of the long average is near optimal.

- The learning curves have a sudden drop at the onset of averaging and are not necessarily informative of their eventual performance before that. Early stopping of seemingly unpromising training runs can cull solutions that would benefit more from averaging.

Motivated by these problems, we propose the two-tailed averaging algorithm with the following features.

- Anytime: An estimate of the optimal tail is maintained at all optimization steps.
- Adaptive: It has no hyperparameters. The number of weights averaged (the length of the tail) is determined adaptively based on the evolution of generalization performance.
- Optimal once in a while: The tail length achieves near optimality regularly.

The algorithm is very easy to implement. Its principal cost is the storage required for a second running average, and it also performs more evaluations of generalization performance (e.g. of the validation loss). The main idea, sketched in Figure 1, is to maintain two running averages of optimization iterates: a *short* and a *long* one, with the long average being our estimate of the optimal weights.

## 2 RELATED WORKS

### 2.1 AVERAGING IN PURE OPTIMIZATION

Polyak averaging as originally proposed (Ruppert, 1988; Polyak and Juditsky, 1992) computes the equally weighted average

$$\bar{\theta}_t = \frac{1}{t+1} \sum_{i=0}^{t} \theta_i$$

of all iterates $\theta_i$ from the optimizer up to the current time step $t$. The convergence rate of $\bar{\theta}_t$ was analyzed in the convex case with an appropriately decaying learning rate. Beyond this strictest interpretation, Polyak (or Polyak-Ruppert) averaging may refer to using $\bar{\theta}_t$ without the convexity assumption, without a decaying learning rate, or with another optimizer such as Adam (Kingma and Ba, 2014).

In practice, where finite budget considerations override the asymptotic optimality guarantees offered by theory, Polyak averaging may refer to an exponential moving average (EMA) of the form

$$\bar{\theta}_0 = \theta_0, \tag{1}$$
$$\bar{\theta}_t = (1 - \beta_t)\theta_t + \beta_t \bar{\theta}_{t-1} \qquad (t \geqslant 1), \tag{2}$$

where $\beta_t < 1$ may be a constant near 1 or it may be scheduled as by Martens (2020). The idea here is to improve the rate of decay of the effect of the initial error by downweighting early iterates.

Tail averaging (TA) (Jain et al., 2018), also known as suffix averaging (Rakhlin et al., 2011), considers a finite optimization budget of $n$ steps with a constant learning rate. At the cost of introducing a hyperparameter $s$ to control the start of averaging, it improves the rate of decay of the effect of the initial error while obtaining near-minimax rates on the variance. Tail averaging is defined as

$$\bar{\theta}_t = \theta_t \qquad (t < s), \tag{3}$$

$$\bar{\theta}_t = \frac{1}{t-s+1} \sum_{i=s}^{t} \theta_i \qquad\qquad (t \geqslant s). \qquad\qquad (4)$$

We have discussed a few representative averaging methods intended for pure optimization but often repurposed in practice for improving generalization by tuning their hyperparameters. Although there are many interesting developments in this area (Shamir and Zhang, 2013; Lacoste-Julien et al., 2012), we now move on to discussing averaging for improving generalization.

## 2.2 AVERAGING FOR GENERALIZATION

In work parallel to tail averaging, Izmailov et al. (2018) propose Stochastic Weight Averaging (SWA), an additional stage of optimization with a constant or cyclical learning rate, that computes an equally weighted average of iterates. SWA can be motivated heuristically in the following way: With the high learning rate, it seeks out wider and flatter basins in the training loss surface to improve generalization, but the high learning rate also prevents it from reaching the bottom of the basin, so the weights bounce around it, thus taking their average should land closer to the minimum point. The SWA algorithm is almost identical to tail averaging (except for a possibly cyclical learning rate and a periodic subsampling of iterates), but it is motivated from the angle of ensembling and generalization.

Our goal is to improve generalization, so the decision of when to start averaging the weights should depend on generalization performance. Indeed, Merity et al. (2017) propose an algorithm much like SWA, where averaging is triggered when the validation loss does not improve for a fixed number of optimization steps, which trades one hyperparameter for another and is sensitive to noise in the raw validation loss.

In other related work, Guo et al. (2022) investigate the repeated application of SWA. Their method is not informed by the validation loss and requires the schedule of multiple SWA stages to be prespecified. Finally, taking the exponential moving average of iterates is also sensitive to its hyperparameter, the decay rate.

In summary, existing averaging methods for generalization which behave well in practice all have one or more hyperparameters to govern the weighting of early iterates. Tuning these hyperparameters can be rather costly, particularly in the presence of other hyperparameters and when training runs take a long time. Furthermore, even with their hyperparameters, these methods are not flexible enough to estimate the optimal average at multiple optimization steps in general. We address these issues in the present work.

The rest of this paper is structured as follows. In Section 3, we formally define the problem to solve. In Section 4, we provide a description of the algorithm, whose properties are analyzed in Section 5. We verify our analysis experimentally in Section 6 and discuss the validity of our assumptions in Section 5.1.

## 3 PROBLEM STATEMENT

Let $\Theta$ be the parameter (or weight) space, $\theta_t \in \Theta$ ($t \in \mathbb{N}_0$) a series of iterates produced by stochastic optimization with $\theta_0$ being the initial value, and $f \colon \Theta \to \mathbb{R}$ the generalization loss function. The generalization loss can simply be the validation loss, or it may measure the classification accuracy while training is minimizing the cross-entropy loss, or it might evaluate on question answering a language model trained with a masked language modelling loss.

We assume the generalization loss is evaluated periodically, every $E \in \mathbb{N}$ optimization steps, and it is at these points $n \in [0, E, 2E, 3E, \dots]$ where we would like to know how many of the most recent iterates to average to minimize it. Here and in the following, $t$ and $n$ (with or without subscripts) are assumed to be from $\mathbb{N}_0$ and $[0, E, 2E, 3E, \dots]$, respectively, and "loss" always refers to $f$.

Defining the average of most recent $\Delta$ iterates up to time step $t$ as $\mathrm{avg}(t, \Delta) = \frac{1}{\Delta} \sum_{i=t+1-\Delta}^{t} \theta_i$, we can define the optimal averaging length as $O(t) = \arg\min_{\Delta \in [1, \dots, t]} f(\mathrm{avg}(t, \Delta))$, and our task is to approximate $O(n)$ at all evaluation steps $n$.

---

**Algorithm 1** Two-tailed averaging with 2 running averages, a short one $\theta^S$ and a long one $\theta^L$ with $S \leqslant L$ number of iterates. If the loss with $\theta^S$ becomes lower or equal than the loss with $\theta^L$, then we switch the long average over to the short one and start a new short average from the empty state. The grayed out parts reset an average if it has not improved for a few evaluations. This heuristic makes the algorithm quicker to adapt when Assumption 4 is violated (see Section 5.1).

---

**Require:** generalization loss function $f$, optimization iterates $\theta_t$
**Ensure:** The sequence $\text{len}_i$ contains how many of the most recent iterates were averaged to compute $\text{loss}_i$ at the $i$th evaluation ($i \in [1, 2, \dots]$).

  1: $S, \theta^S, L, \theta^L \leftarrow 0, \mathbf{0}, 0, \mathbf{0}$                    ▷ The first evaluation after this init will cause a switch.
  2: **for** $t \leftarrow 1, 2, \dots$ **do**
  3:      $S, \theta^S \leftarrow S + 1, \frac{S\theta^S + \theta_t}{S+1}$                           ▷ Add $\theta_t$ to the short average
  4:      $L, \theta^L \leftarrow L + 1, \frac{L\theta^L + \theta_t}{L+1}$                           ▷ Add $\theta_t$ to the long average
  5:      **if** $t \bmod E = 0$ **then**                      ▷ Evaluate the loss every $E$ iterates
  6:          $F^1, F^S, F^L \leftarrow f(\theta_t), f(\theta^S), f(\theta^L)$
  7:          **if** $F^S$ is deteriorating **then**
  8:              $S, F^S \leftarrow 0, +\infty$                       ▷ Reset deteriorating $S$
  9:          **end if**
10:          **if** $F^S \leqslant F^L$ or $F^L$ is deteriorating **then**       ▷ Is the short average better?
11:              $S, L, \theta^L, F^L \leftarrow 0, S, \theta^S, F^S$      ▷ Switch: set $L$ to $S$ and reset $S$
12:          **end if**
13:          $i \leftarrow t/E$
14:          **if** $F^1 \leqslant F^L$ **then**                   ▷ Is the current iterate better?
15:              $S, L \leftarrow 0, 0$                        ▷ Reset both averages
16:              $\text{loss}_i, \text{len}_i \leftarrow F^1, 1$        ▷ Report the loss and averaging length 1
17:          **else**
18:              $\text{loss}_i, \text{len}_i \leftarrow F^L, L$         ▷ The long average is better, report it
19:          **end if**
20:      **end if**
21: **end for**

---

There is a trivial algorithm to find $O(n)$, that saves all $\theta_i$ and performs a search over $\Delta \in [1, \dots, n]$ to minimize $f(\text{avg}(n, \Delta))$. This has storage and evaluation cost proportional to $n$, which is prohibitive. Even assuming that $f$ improves monotonically in $\Delta$ up to its optimum, the cost is still proportional to $O(n)$. Our proposed algorithm approximates $O(n)$ with a constant cost.

## 4   The Algorithm

Algorithm 1 specifies two-tailed averaging (TTA) in pseudocode. The main algorithm (without the grayed out bits) works as follows. The outer loop iterates over weights $\theta_t$ produced by a stochastic optimizer, incorporating them into the short and long running averages $\theta^S$, $\theta^L$ with lengths $S$ and $L$. Then, every $E$ steps, the loss is evaluated with the current weights $\theta_t$, with the short average $\theta^S$, and with the long average $\theta^L$, giving $F^1$, $F^S$ and $F^L$. If $F^S$ is better than $F^L$, then we switch: the long average continues from the current short average and the short average is restarted (see Figure 1). After this, we know that the long average is strictly better than the short one, but $F^1$ might be better still if the raw loss is improving fast, which is the case very early in training. In any case, we report the best of $F^1$ and $F^L$ for that evaluation.

The grayed out logic constitutes the so-called reset heuristic. By resetting the short and long averages if they haven't improved for a number of evaluations, it is intended to handle cases where the averages become too long, perhaps due to optimization escaping from one basin of attraction to a better one or to the loss surface changing in a non-stationary environment.

This algorithm is online as it only accesses the current weights. We analyze its other properties in the next section.

## 5 ANALYSIS OF THE ALGORITHM

Our analysis hinges on the following assumptions, which follow from, for example, a monotonically decreasing loss and averaging producing diminishing returns as the length increases, and whose validity is discussed in Section 5.1.

**Assumption 1.** *For all $n$, as a function of $\Delta \in [0, E, 2E, \ldots, O_E(n)]$, $f(\mathrm{avg}(n, \Delta))$ is monotonically decreasing, where $O_E(n) = \lfloor O(n)/E \rfloor E$. That is, for any given evaluation step $n$, averaging over more iterates from the past monotonically improves $f$ until about the optimum length.*

**Assumption 2.** *For all $n$ and all $n_+ \geqslant O^E(n)$, $f(\mathrm{avg}(n, O^E(n))) \leqslant f(\mathrm{avg}(n, n_+))$, where $O^E(n) = \lceil O(n)/E \rceil E$. That is, averaging slighly more iterates than optimal is better than averaging a lot more.*

**Assumption 3.** $\forall n \colon \exists n_s \colon O(n + n_s) - O(n) < n_s$, *that is, the optimal average drops some iterates over a sufficiently long interval.*

**Assumption 4.** $O(n) \leqslant O(n + E))$, *that is, the optimal number of weights to average is monotonically increasing from one evaluation to the next.*

Let $S(t)$, $\theta^S(t)$, $L(t)$, and $\theta^L(t)$ stand for the values of variables $S$, $\theta^S$, $L$, and $\theta^L$ in Algorithm 1, respectively, after $t$ times through the outer loop. Similarly, let $S'(t)$, $\theta^{S'}(t)$, $L'(t)$, and $\theta^{L'}(t)$ stand for the values of the same variables at the same iteration but after Line 4 (i.e. before the possible switch of the short and long averages). Furthermore, we introduce the shorthands $f^X(t) = f(\mathrm{avg}(t, X(t)))$ for $X \in \{S, S', L, L', O\}$ with $f^X(t) = +\infty$ if $X(t) = 0$.

**Definition 1** (Switch point). *We say that $n$ is a switch point if at $t = n$ Line 11 is executed, that is, when the short average becomes at least as good as the long average with respect to the loss, and consequently the short average is made the long average while a new short average is started. We denote the most recent switch point before iteration $t$ with $\mathrm{SP}(t)$, where $\mathrm{SP}(t) < t$. If there is no such switch point, then $\mathrm{SP}(t) = -1$.*

Assumption 4 states that the optimal averaging length monotonically increases, so without loss of generality, to simplify the analysis, we assume throughout that the raw loss $F^1$ has already been eclipsed by $F^L$ at the first evaluation. For the analysis, we also assume that the reset heuristic (the grayed out parts in Algorithm 1) cannot trigger.

**Proposition 2** (Basic properties). $\forall t \geqslant E \colon$ *and* $\forall n \geqslant E$:

  (i) $E \mid S(n), E \mid L(n)$            *(averaging lengths are multiples of the evaluation period)*
  (ii) $S(n) \leqslant L(n) \leqslant n$    *(because they only increase except at switches where S is reset to 0)*
  (iii) $f^S(n) > f^L(n)$                            *(due to the switching logic)*
  (iv) $L(t) = S(t) + S'(\mathrm{SP}(t))$ *if* $\mathrm{SP}(t) \neq -1$ *else* $L(t) = S(t)$     *(due to the switching logic)*

**Proposition 3** (Bounds for the averaging lengths). *The lengths of the short and long averages are bounded as $S(n) < O(n)$ and $L(n) < 2O(n) + E$.*

*Proof.* We prove $S(n) < O(n)$ by contradiction. Suppose $O(n_0) \leqslant S(n_0)$ for some $n_0$. As $O(n)$ is monotonically increasing and $S(n)$ increases by $E$, there exists $n \leqslant n_0$ such that $O^E(n) = S(n)$ (Proposition 2). Since $S(n) \leqslant L(n)$ (by (ii) of Proposition 2), from Assumption 2 and $O^E(n) = S(n) \leqslant L(n)$, we have that $f^S(n) \leqslant f^L(n)$, which contradicts (iii) of Proposition 2.

Next we prove $L(n) < 2O(n) + E$. From (iv) of Proposition 2, we have that at the beginning, when there has not yet been a switch, $L(t) = S(t)$; else $L(t) = S(t) + S'(\mathrm{SP}(t))$ for all $t$. In the first case, $L(n) = S(t) < O(n) < 2O(n) + E$, and we are done.

In the second, the usual case, $L(n) = S(n) + S'(\mathrm{SP}(n))$. That is, the length of the current long average is the sum of the lengths of the current and the previously finished short average. Since $S(n) < O(n)$ and $S'(SP(n)) = S(SP(n) - E) + E$, so $L(n) < O(n) + S(SP(n) - E) + E$, from which $L(n) < O(n) + O(\mathrm{SP}(n) - E) + E$. Finally, from the monotonicity of $O$ in Assumption 4, $O(\mathrm{SP}(n) - E) \leqslant O(n)$, we get $L(n) < 2O(n) + E$. $\qquad\square$

**Proposition 4** (Infinite number of switch points). *Switch points keep coming, that is, $\forall n \colon \exists n_s \geqslant n \colon SP(n_s) \neq -1$.*

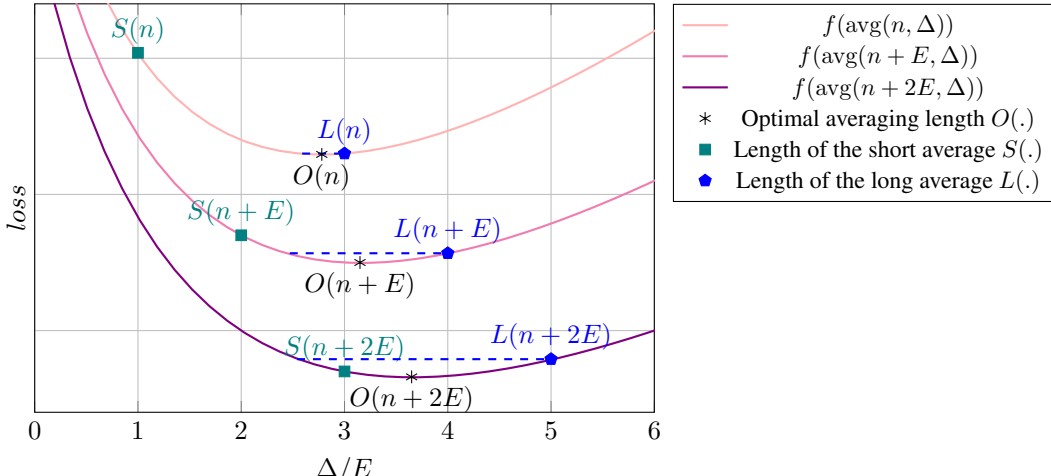

Figure 2: Idealized illustration of switching. The three curves show the loss as a function of averaging length at three subsequent evaluations (at optimization steps $n$, $n + E$, and $n + 2E$). The raw loss keeps improving, hence later evaluations have lower loss curves. The optimal averaging length increases, $O(n) \leqslant O(n + E) \leqslant O(n + 2E)$, as per Assumption 4. At $t = n + 2E$, where the loss of the short average dips below the loss of the long average, the long average is discarded, the short average becomes the long average, and a new short average is started with zero length.

*Proof.* This follows from $S(n) < O(n)$ and that $S(n)$ increases by $E$ between switch points, eventually catching $O(n)$, that grows more slowly by Assumption 3. $\qquad\square$

**Proposition 5** (Once-in-a-while optimality). *While between any two subsequent switch points $n_1$ and $n_2$, the long average is nearly optimal at least once. Formally, $\exists n \in [n_1, n_2]: L(n) = O^E(n)$.*

*Proof.* Since $S(n) < O(n)$ and $E \mid S(n)$ for all $n$, so $S'(n_1) \leqslant O^E(n_1)$. Then it is either that $S'(n_1) = O^E(n_1)$ or $S'(n_1) < O(n_1)$. Since at switch points $L(n) = S'(n)$, in the former case, we conclude the proof with $n = n_1$. Considering the latter case, $L(n_1) = S'(n_1) < O(n_1)$. Also, switches happen when $f^{S'}(n) \leqslant f^{L'}(n)$, but as per Assumption 1 this can happen only if $O^E(n) \leqslant L(n)$. Thus for $n_2$ to be a switch point, it must be that $O(n_2) \leqslant L(n_2)$, and we get that $L(n_1) < O(n_1) \leqslant O(n_2) \leqslant L(n_2)$. Therefore, considering that $L(n)$ increases linearly between switch points, there must be a point $n$ where $L$ becomes greater than or equal to $O$, and at that point $n = O^E(n)$ because $E \mid L(n)$ and $O$ increases monotonically according to Assumption 4. $\qquad\square$

In short, we have shown that there is an infinite number of switch points, the long average cannot be more than about twice as long as optimal, and between any two switch points the long average gets as close to the optimal length as possible given the periodic evaluation scheme. Our results are in terms of lengths of averages, and relating the actual loss with the long average $f^L$ to the loss with the optimal length $f^O$ would be desirable. Here, we informally point out that, all things being equal, the worse $f^L$ gets relative to $f^O$, the quicker $f^S$ is to catch up, making long periods of highly suboptimal solution less likely. Formalizing this notion requires making further assumptions about the loss-vs-averaging-length function (of the kind plotted in Figure 2) and would make analysis considerably more cumbersome.

## 5.1 ON FAILED ASSUMPTIONS

To augment the theoretical analysis, that is based on idealized assumptions, we make the following observations. The strongest assumption by far is Assumption 1. It assumes that increasing the averaging length monotonically improves $\hat{f}$ until the optimum. In terms of the loss, the algorithm is fairly robust to when the assumption holds only approximately because small deviations of $f(\text{avg}(n, \Delta))$ from monotonicity can change switch times only when $f^S$ and $f^L$ are close.

Assumption 2 says that averaging slightly more iterates than optimal (i.e. rounded up to the evaluation period) is better than averaging a lot more. This is a weak assumption. If it fails, the algorithm can

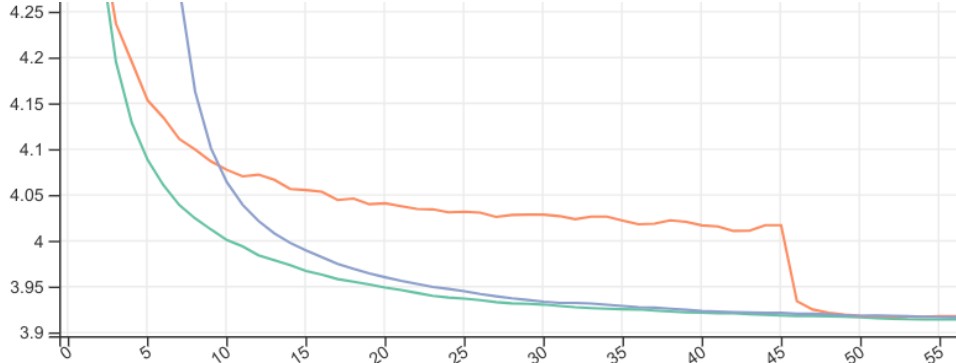

Figure 3: Validation loss with two-tailed averaging (TTA, green), tail averaging (TA, orange), and an exponential moving average (EMA, blue) of weights on language modelling on Penn Treebank. For both TA and EMA, their hyperparameters (the start time and the decay rate) were tuned to minimize the final loss, so it is not a surprise that all three have similar optima. TTA has no hyperparameters and produces much better early solutions. These two factors make tuning easier and early stopping much more reliable. Additionally, the noise in the raw loss is effectively smoothed out. Note that while the TTA loss decreases monotonically, gentler and steeper slopes are manifest before and after switch points, respectively.

fail to detect when the short averages becomes longer than optimal, which delays switching until the next evaluation.

Assumption 3 failing means that the optimal average incorporates all new iterates without ever dropping old ones. In this case, the short average, which is always shorter than optimal, will be a constant number of iterates behind and its loss will converge to the loss of the optimal average. If the long average is shorter than optimal, then the same argument applies to it. If the long average is longer than optimal, then eventually a switch will happen. In either case, the loss of the long average converges to that of the optimal average.

Regarding Assumption 4, $O(n) \leqslant O(n + E)$ can fail if, for example, the improvement of the raw loss accelerates, but that is a rather uncommon and temporary occurrence. It may also fail if the raw loss has started to worsen due to overfitting or optimization has escaped from one basin to the next and the average is slowly climbing the ridge separating them. Yet another way for this assumption to fail is when the loss landscape changes during learning in a non-stationary environment. With the exception of accelerating improvement, these cases are likely to be caught by the reset heuristic, wherein the short and long averages are reset if their loss does not improve for a few evaluations (see the grayed out parts in Algorithm 1).

The reset heuristic can trigger when it shouldn't, i.e. when Assumption 4 holds. Such a spurious reset makes the estimate of the long average worse either directly or indirectly by delaying the next switch. Either way, without further violations of this assumption, the algorithm recovers by the next switch point.

All in all, we can expect the algorithm to display some degree of robustness to minor violations of the assumptions. In practice, we recommend choosing a reasonably large $E$ to reduce the noise originating from the stochasticity of optimization and the likelihood of the resulting violations.

## 6 EXPERIMENTS

Tail averaging or Stochastic Weight Averaging have been shown previously to be beneficial not only in theory and on simulated data (Jain et al., 2018) but also in language modelling (Merity et al., 2017; Melis et al., 2019) and image classification (Izmailov et al., 2018) experiments. Hence, in this work we restrict our attention to experiments in a single domain to corroborate the analysis in Section 5. Our goals are to *i)* verify that TTA is on-par with well-tuned TA and EMA, *ii)* explore the effect of basing the switching logic on the training loss instead of the validation loss, *iii)* demonstrate robustness to the choice of evaluation period, *iv)* and check how well the assumptions made in Section 5 hold in practice.

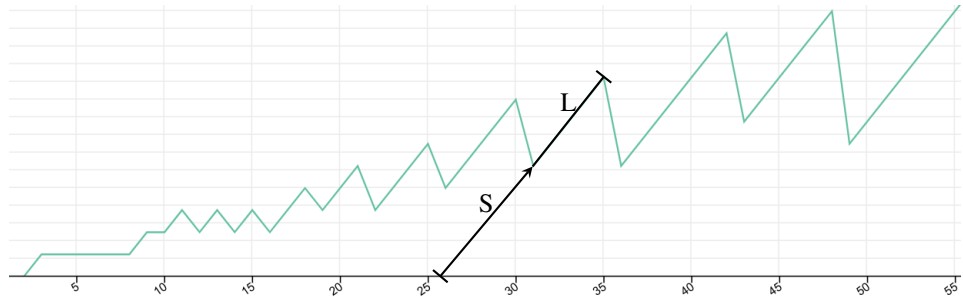

Figure 4: Length of the long average ($L$) vs number of evaluations of TTA in Figure 3. Note how the heights of both peaks and valleys (corresponding to $L(n)$ and $S'(n)$) increase almost monotonically. Also, the correspondence between this figure and the schematic in Figure 1 is illustrated on one of the $S$ and $L$ intervals.

In particular, we trained a recurrent language model with several hyperparameters on Penn Treebank (Mikolov et al., 2010) using the Rectified Adam optimizer (Liu et al., 2019), with periodic evaluation every 1000 optimization steps. The hyperparameters were tuned separately for TTA, TA eq. (3), and EMA eq. (1). Figure 3 shows that the final validation losses with the three methods are very close, but early losses with TTA are much better. This is expected because TA and EMA are not flexible enough to produce optimal averaging lengths at multiple points along the learning curve despite having an extra hyperparameter. Conversely, TTA has at least one nearly optimal solution between any two subsequent peaks (i.e. switch points) in Figure 4 despite having no hyperparameters.

We also tried the version of Algorithm 1 where the switching logic was based on comparison of the training losses of the short and long averages instead of the validation losses, but the true validation losses were reported. On this particular language modelling task, the best validation loss with the modified algorithm worsened moderately (3.93 vs 3.92) and was well below the raw validation loss (4.02). Results on the test set exhibited the same gap. Since the modified TTA was minimizing the training loss, the smoothness of the reported validation losses observed in Figure 3 were lost in the process. Similar results were obtained by scheduling a learning rate drop without iterate averaging.

To explore the effect of the evaluation period $E$, we tuned models with four times larger and four times smaller $E$ than in our previously discussed experiments. As expected, the best final results were very close to each other, with shorter periods having an advantage early in training as the raw loss $F^1$ was more quickly eclipsed by $F^L$.

In addition, we found that the assumptions made in Section 5 held rather well in practice: the resulting loss $F^L$ in Figure 3 and the averaging lengths tended to change monotonically (see heights of peaks and valleys in Figure 4), making our length-based theoretical results more closely linked to the actual loss. When that was not the case, we found that the raw loss $F^1$ had started to worsen due to overfitting or, much more rarely, optimization had entered a new basin, violating Assumption 4. Figure 5 and Figure 6 demonstrate the reset heuristic being triggered in these cases. Finally, TA and TTA having almost identical final validation losses weakly supports our assumptions, although a conclusive demonstration would need to plot the results obtained with TA tuned separately for each evaluation, which is beyond our computational budget.

## 7 CONCLUSION

In summary, we presented a variant of tail averaging and Stochastic Weight Averaging based on two running averages. Compared to them, two-tailed averaging requires additional storage for the second running average and relies on periodic evaluation of generalization performance. In return, TTA both removes a hyperparameter and provides an estimate of the optimal tail at all optimization steps. In effect, it makes hyperparameter tuning easier and early evaluation much more representative of final performance, allowing it to support early and anytime stopping better. Owing to its simplicity, low implementation cost and adaptivity, TTA is a practical and widely applicable method for improving generalization.

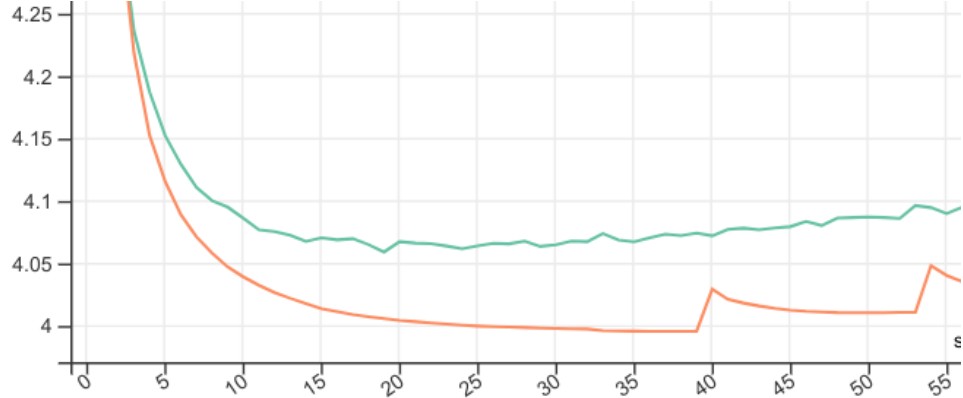

Figure 5: Raw (green) and TTA (orange) validation loss with overfitting. The averaged loss bottoms out due to overfitting. Thus when the averages are reset (twice), the validation loss does not recover. Although the losses reported after the reset are suboptimal, it does not really matter as a better loss was reported already.

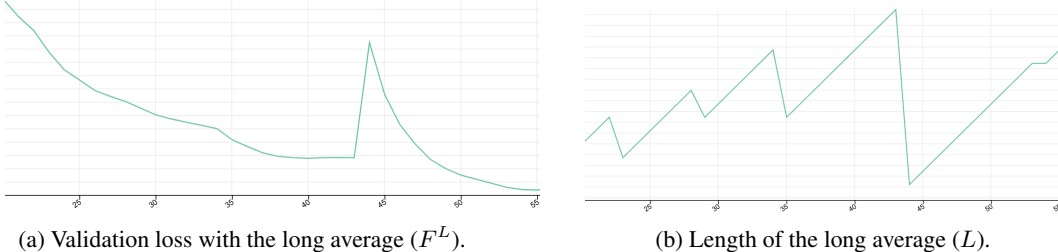

(a) Validation loss with the long average ($F^L$).    (b) Length of the long average ($L$).

Figure 6: Example of optimization entering a new basin. At $t = 40E$ the validation loss bottoms out and starts to slightly worsen. This is detected at $t = 43E$, and both averages are reset. With the averages now way too short, the loss spikes but then recovers.

Looking beyond the scope of this work, exploring the relationship between iterate averaging and learning rate schedules is a promising direction as existing (Merity et al., 2017) and our own limited experimental results indicate that dropping the learning rate and tail averaging perform comparably.

The properties of our algorithm are particularly compelling for continual learning: by allowing the learning rate to remain high and being able to adapt the averaging length to changing circumstances, TTA lets the model maintain high plasticity while reaping the benefits of averaging.

In addition, averaging weights can be viewed as a cheap approximation to averaging predictions when the averaged weights reside in a region with a suitable geometry. Averaging weights within such regions and averaging predictions over regions (each with its own weight average) could potentially achieve a better loss than weight averaging alone at much lower storage and evaluation cost than pure prediction averaging. We leave these avenues for future work to explore.

## 8 REPRODUCIBILITY

Our main contributions, Algorithm 1 and its analysis are fully contained in the paper.

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
