# OpenReview forum: "Two-Tailed Averaging: Anytime Adaptive Once-in-a-while Optimal Iterate Averaging for Stochastic Optimization"
_ICLR.cc/2023/Conference — Submitted to ICLR 2023_

### Official Review · Reviewer_bawy · 2022-10-21

**Confidence:** 3
**Correctness:** 4
**Technical Novelty And Significance:** 4
**Empirical Novelty And Significance:** 4
**Recommendation:** 8

**Clarity, Quality, Novelty And Reproducibility:**

Clarity: The paper is written clearly and the analysis is elegant and understandable

Quality: The paper presents a high quality theoretical analysis, the experiments section could be improved by being a bit more extensive

Novelty: I'm not familiar with the tail averaging literature so I'm not sure about this point but to me the presented method seems like novel and original work

Reproducability: The focus of the paper is on the algorithm and theory which is clearly presented and reproducable. The presented experiment is a typical setup which could be easily reproduced with simila
r models and the same dataset which is publicly available.

**Strength And Weaknesses:**

Strength:
- Paper does a great job explaining the algorithm itself and also the motivation for it
- Assumptions are clearly stated and even though they seem to be too strict for a nonconvex setting they should still be motivated in practice close to optimum and otherwise treated by the heuristics
- Great figures, they take some time to think about but then make sense and illustrate useful properties once you understand them
- Figure 3 nicely demonstrates that the approach competes with hyperparameter tuned alternatives but in a purely adaptive fashion
- Figure 5 nicely demonstrates the TTA has the potential to improve over not using any tail averaging
- Figure 6 nicely demonstrates the effect of a required reset heuristic from nonconvexity and again how much too-short-averaging-length hurts the validation loss

Weakness:
- Authors fairly point out that the analysis would become more cumbersome when trying to make a statement about the worst case difference between f^O and f^L but explain why this should not be a problem in practice. Under simplified assumptions where f is convex extending the analysis could potentially not be as cumbersome? Perhaps the loss is also monotonous in the averaging length before the averaging length becomes suboptimal? Having this analysis would present a nice addition for the paper making the theoretical analysis even more high quality.
- Figure quality (e.g. of Figure 3, Figure 5) could be improved (the numbers on the axes look blury)
- Would be nice to see the experimental evaluation also on modern architectures and in general on a broader set of evaluations
- The structure of the experiments section could be clearer by separating the different research questions into separate sections making the presentation cleaner

Questions:
- How much wall-clock time cost is added in practice by maintaining the additional sets of weights?
- In this paper about anytime tail averaging: https://arxiv.org/pdf/1902.05083.pdf the averaging length k seems to depend on t (k=ct) which makes maintaining the average more expensive and still requires tuning c. But such approach does not seem to be used for comparison in the author's paper / they seem to compare to a constant averaging length "only". Does the k=ct approach behave qualitatively differen
tly / more similar to the author's proposed adaptive method before the last iterate for which it was tuned?
- The authors mention that tail averaging performs similarly to learning rate decay. Are there comparable adaptive methods for learning rate decay and how does the presented method compare in terms of com
putational cost (memory and wall-clock time)?

**Summary Of The Paper:**

The authors propose a tail averaging method for stochastic optimization that maintains two averages of different lengths in order to adaptively choose tail averging length.
The average produced by the longer of the two averages is provably close to optimal in length every once in a while.
The advantage of the proposed method over existing tail averaging methods is that it does not have tail averaging length as a hyperparameter.
The only hyperparameter is the number of iterations that pass between evaluating and comparing the two averages and the proposed method is robust to larger changes of this parameter.
The paper provides a theoretical analysis of the algorithm under a set of assumptions and argues how applicable the assumptions are in practice.
The paper also provides a demonstration that the method works well in practice also compared to other iterate averaging approaches for a toy example using an LM on Penn Treebank.

**Summary Of The Review:**

I recommend to accept the paper due to the clear and elegant theoretical contribution that adds to the tail averaging literature, an important area of stochastic optimization, and the method is also directly applicable and practically feasible for many different machine learning applications.

---

> ### Author Response · Authors · 2022-11-18
> **Response to bawy**
>
> Thank you for the detailed review.
> See our responses below.
>
> > Authors fairly point out that the analysis would become more cumbersome when trying to make a statement about the worst case difference between f^O and f^L but explain why this should not be a problem in practice. Under simplified assumptions where f is convex extending the analysis could potentially not be as cumbersome? Perhaps the loss is also monotonous in the averaging length before the averaging length becomes suboptimal? Having this analysis would present a nice addition for the paper making the theoretical analysis even more high quality.
>
> Yes, this would be a nice addition to the paper. To provide some background, our motivation was improving generalization performance, hence pure optimization considerations were neglected. The focus of the paper is still very much on generalization, and we updated the title, abstract and the text to emphasize that. See *Focus on Generalization* in our response to Reviewer bYZd.
>
> > Would be nice to see the experimental evaluation also on modern architectures and in general on a broader set of evaluations
>
> We kind of waved this away by saying that many previous works showed averaging to be useful, but as Reviewer NMLB points out, this carries over to TTA in only as much as our assumptions for the analysis of the algorithm hold.
>
> > How much wall-clock time cost is added in practice by maintaining the additional sets of weights?
>
> Maintaining an additional set of weights was negligible in our experiments. The main cost there in practice is moving the weights off the GPU/TPU, which is the same with one or two running averages. Thus for very large models, we recommend down-sampling the iterates to be averaged and updating the moving averages on disk (easy, only needs sequential access).
>
> > In this paper about anytime tail averaging: https://arxiv.org/pdf/1902.05083.pdf the averaging length k seems to depend on t (k=ct) which makes maintaining the average more expensive and still requires tuning c. But such approach does not seem to be used for comparison in the author's paper / they seem to compare to a constant averaging length "only". Does the k=ct approach behave qualitatively differently / more similar to the author's proposed adaptive method before the last iterate for which it was tuned?
>
> With tail averaging, the cost of computing the average of length $ct$ at *all* time steps t has a prohibitive storage cost, and a higher computational cost. TTA solves this efficiency problem (and makes $c$ adaptive). We added a paragraph on costs to "Problem Statement". Our experimental comparison is meant to reflect how in practice these methods can be used with reasonable computational cost.
>
> > The authors mention that tail averaging performs similarly to learning rate decay. Are there comparable adaptive methods for learning rate decay and how does the presented method compare in terms of computational cost (memory and wall-clock time)?
>
> As discussed above, TTA (and averaging in general) does have some overhead over tweaking the learning rate. The effect of high learning rates on generalization may prove to be an important advantage of averaging. As to adaptive learning rates, often the learning rate is dropped when validation performance starts to plateau, whose averaging counterpart is Stephen Merity's (2017) non-monotonically triggered ASGD. We are not aware of fancier adaptive learning rate schedulers.

---

### Official Review · Reviewer_NMLB · 2022-10-25

**Confidence:** 4
**Correctness:** 3
**Technical Novelty And Significance:** 2
**Empirical Novelty And Significance:** 2
**Recommendation:** 3

**Clarity, Quality, Novelty And Reproducibility:**

**Clarity**: The paper is generally well written. Some of theoretical derivations were
difficult to follow because of poor notation choices (see "Minor Comments").

**Reproducibility**: The experiments are not reproducible with the information
provided. Batch-size and other critical details are missing for the experiments in Figures 3/4.

**Missing References**:
- Tail averaging as described by Jain et al. (2018) is functionally identical to
    suffix averaging, which was proposed by Rakhlin et al. [1] in 2012.
    Rakhlin et al. proved that suffix averaging is optimal for strongly convex
    stochastic optimization. I'm not certain why Jain et al. don't discuss
    the work by Rahklin et al., given that they even note tail averaging is
    the same method on page 4 of their work.
    Regardless, this highly related branch of work should be properly discussed
    here.

- Several other works have developed anytime variants of suffix averaging which
    are also optimal. For example, polynomial-decay averaging [2] is easy to compute
    and also obtains the optimal convergence rate. Lacoste-Julien et al. [3]
    provide yet another optimal anytime averaging scheme with a simpler proof
    of convergence than that for polynomial-decay averaging.

**Novelty**: The two-tailed averaging method is new to the literature as far as I am aware.

### Minor Comments

- Equation 2: Note that Polyak-Rupert averaging is easily implemented as an EMA using
    $\beta_t = t/(t+1)$. The optimal scheme of Lacoste-Julien et al. [3] is also easily
    implemented in this fashion.

**Related Works**:
- "In summary, existing averaging methods which behave well in practice all have one or more hyper- parameters to govern the weighting of early iterates." --- this is not necessarily true. See "Missing References".
- "these methods are not flexible enough to estimate the optimal average at multiple optimization steps in general" --- this is not true once the missing references are considered.

**Problem Statement**:
- The notation $O(t)$ for optimal averaging length is somewhat confusing due to the clash with standard "Big-O" notation.

**Analysis**:
- Given assumption 3, doesn't assumption 4 simply imply the conclusion holds for all $E$? I don't see why the precondition is even introduced here given assumption 3.
- Figure 2: Is this generated from a real optimization problem, or is it just a synthetic example?
- What is the notation $E | S(n)$ in Prop. 2 (i)? It doesn't appear to be introduced anywhere.

### References

[1] Rakhlin, Alexander, Ohad Shamir, and Karthik Sridharan. "Making gradient descent optimal for strongly convex stochastic optimization." arXiv preprint arXiv:1109.5647 (2011).

[2] Shamir, Ohad, and Tong Zhang. "Stochastic gradient descent for non-smooth optimization: Convergence results and optimal averaging schemes." International conference on machine learning. PMLR, 2013.

[3] Lacoste-Julien, Simon, Mark Schmidt, and Francis Bach. "A simpler approach to obtaining an O (1/t) convergence rate for the projected stochastic subgradient method." arXiv preprint arXiv:1212.2002 (2012).

**Strength And Weaknesses:**

The main strength of the submission is the relative simplicity and elegance of
the averaging method.
By maintaining two averages --- one over a short time scale and one over a long time scale ---
the averaging scheme can capture both local improvement in the objective
and long-term behavior of the optimizer.
I think it is possible that the length of the short/long intervals could be
set to give near-optimal anytime convergence rates for the training objective,
e.g. something like Theorem 5 of Rakhlin et al. [1].

The major weaknesses of the submission are the following: (i) the theoretical
justification for two-tailed averaging is unrealistic and
(ii) the experimental evaluation is to small in scope to justify the
method without strong theory.

**Theoretical Justification**: The theory in this submission does not justify
the proposed method.
My major complaint here is that the authors assume exactly the properties of
the validation loss that they need for their averaging scheme to work.
In particular, Assumptions 1 --- monotonic behavior of the validation loss
with respect to the length of the average --- is deeply unrealistic for several
reasons:

- It is unlikely to be satisfied for stochastic optimization of non-convex models,
    where monotone improvement of any train/validation/test metric is unlikely.
- It makes a very strong assumption on the connection between the training objective
    and the validation objective. That is, progress on the training objective
    should roughly equate to progress on the validation metric.

Because of the weakness of the theoretical analysis, I can only evaluate
two-tailed averaging as a heuristic method for setting the suffix parameter.

**Empirical Evaluation**: The empirical evaluation is small scale and does
not investigate the assumptions on which the theory is based.
Only two experiments are presented, both on Penn treebank.
No comparison is made against other anytime averaging methods
(see "Missing References"), making it difficult to establish a baseline for
the performance of two-tailed averaging.

While two-tailed averaging performs well on Penn Treebank, one experiment does
not justify it's wide-spread use.
In particular, I disagree with the authors statement that two-tailed averaging "is approximately equivalent
to a series of TA which were tuned for each evaluation step separately" and thus requires no other experiments;
this only holds under their assumptions.
While the assumptions were tested via downstream metrics like behavior
of the average lengths, this is only (approximately) addresses Assumption 3.
Checking the assumptions properly requires carefully plotting the behavior of
the validation loss and optimal average length on multiple heterogeneous datasets.

Finally, some experiments appear to be omitted from the manuscript.
Specifically, the authors say: "To explore the effect of the evaluation period E, we tuned models with four times larger and four times smaller E than in our previously discussed experiments."
I do not see these results presented anywhere, so we can only take the authors
on their word that $E$ doesn't strongly affect performance.

**Summary Of The Paper:**

This submission proposes a new iterative averaging scheme for stochastic
gradient methods.
The scheme, which works by maintaining two iterate averages
at any given time, manages to be parameter free by setting the average length
parameters dynamically based on the validation loss.
As such, it is parameter-free unlike suffix/tail averaging, where the tail
length parameter must be specified a-priori.
The authors prove that their "two-tailed" averaging scheme is optimal at
specific iterates infinitely often ("once in a while") under assumptions
on the behavior of the validation loss with respect to the length of the
average.
The submission concludes with several experiments on the Penn Treebank dataset.

**Summary Of The Review:**

=== Update ===

I appreciate the discussion with the authors, who have clarified several issues in the manuscript. I think the paper is improved, but I remain skeptical of the main assumptions required for the theoretical analysis. In particular, I think that Assumption 1 is not justified and I am reluctant to accept this submission without (i) an in-depth experimental justification for the assumption, or (ii) a revised theoretical analysis. Although I would normally increase my score to four reflect our discussion, I cannot due to the review form. As such, I will maintain my score of 3.

=====

This submission is an interesting attempt to develop anytime methods for iterate
averaging. I like two-tailed averaging scheme; perhaps it can be shown to admit
a near optimal convergence rate for the training loss.
Instead of pursing this direction, the authors rely on difficult-to-trust
assumptions on the validation for their theoretical analysis, which shows that the averaging
scheme is once-in-a-while optimal.
As the experiments are limited to a single dataset and do not verify the assumptions,
I feel the proposed method is not well-justified by theory or experiment.
I recommend the submission be rejected.

---

> ### Author Response · Authors · 2022-11-18
> **Response to NMLB**
>
> Thank you for the in-depth review; we incorporated much of what has been suggested.
> See our response to specific issues below, but most importantly see *Focus on Generalization* in our response to Reviewer bYZd.
> In a nutshell, the original submission falsely gave the impression of a normal optimization paper, while it is about generalization.
> We changed quite a bit to reflect this.
>
> > "authors assume exactly the properties of the validation loss that they need for their averaging scheme to work"
>
> Yes, we characterize conditions for the algorithm to have guarantees.
> By stating these conditions explicitly, we invite discussion of their validity.
> Furthermore, we provide some discussion of what happens when the assumptions are broken in Section 5.1 "On Failed Assumptions".
>
> > Assumption 1 is unlikely to be satisfied for stochastic optimization of non-convex models, where monotone improvement of any train/validation/test metric is unlikely.
>
> First, let's consider the training loss (which is *not* our focus at all).
> By training loss, we mean the loss of the model on the training set, not the noisy per-iterate estimate of it that drives SGD.
> We argue that the training loss is very likely to improve monotonically with a not too small period for evaluation.
> If the training loss does not improve for an entire period, then there is a problem with the underlying optimization (or we reached convergence).
> Also, we discuss in section 5.1 how the impact of small deviations from monotonicity is limited.
>
> In the updated paper, we relaxed Assumption 1 to reflect the fact that the proofs only require improvement in the generalization loss f from one evaluation to the next.
>
> For the validation/test metric see our answer to the next comment.
>
> > Assumption 1 makes a very strong assumption on the connection between the training objective and the validation objective. That is, progress on the training objective should roughly equate to progress on the validation metric.
>
> Yes, this assumption is of course there, but it is not our invention: the underlying optimization problem (i.e. without averaging) must be set up reasonably to have any hope of generalizing from the training set to validation loss or good downstream performance.
> More abstractly, this is an assumption of Empirical Risk Minimization (when considering the validation loss).
> If this assumption does not hold, then the learning problem is badly posed, and how to do averaging is the least of our problems.
>
> > The experiments are not reproducible with the information provided. Batch-size and other critical details are missing for the experiments in Figures 3/4.
>
> The experimental details are described in a published paper, which we didn't cite to better preserve anonymity.
> We couldn't provide a complete description either for the same reason.
> We will update the paper for the camera ready ...
>
> > The empirical evaluation is small scale and does not investigate the assumptions on which the theory is based.
>
> We agree that that more experiments would strenghten the case.
> At present, we have indirect support for the assumptions in the last paragraph of Section "Experiments".
>
> > I disagree with the authors statement that two-tailed averaging "is approximately equivalent to a series of TA which were tuned for each evaluation step separately" and thus requires no other experiments; this only holds under their assumptions.
>
> True. We removed this sentence.
>
> > Add missing references.
>
> Done.
>
> > - Given assumption 3, doesn't assumption 4 simply imply the conclusion holds for all? I don't see why the precondition is even introduced here given assumption 3.
>
> It was intended to cover the case when Assumption 3 is broken. It was indeed confusing in its misguided pedantry, we removed the assumption and simply say that we analyse the case where the reset heuristic does not trigger.
>
> > Figure 2: Is this generated from a real optimization problem, or is it just a synthetic example?
>
> It's a purely illustrative, mock example. Updated the caption to reflect this.
>
> > What is the notation  in Prop. 2 (i)? It doesn't appear to be introduced anywhere.
>
> The vertical bar stands for divisibility. Updated the text.

---

> > ### Comment · Reviewer_NMLB · 2022-11-18
> > **Reply to Author Response**
> >
> > Thanks for your response and for addressing many of my comments. I have several further points I would like to make.
> >
> > > Yes, we characterize conditions for the algorithm to have guarantees. By stating these conditions explicitly, we invite discussion of their validity. Furthermore, we provide some discussion of what happens when the assumptions are broken in Section 5.1 "On Failed Assumptions".
> >
> > There is a difference between characterizing the function class on which a method works (e.g. Lipschitz continuous functions) and assuming properties which cannot be verified (e.g. monotonicity of the validation loss) on a specific class of functions. It is this latter issue that I am criticizing. I think your assumptions are akin to assuming that gradient descent makes progress on $f$, instead of assuming $f$ is Lipschitz smooth and then deriving a progress condition.
> >
> > > First, let's consider the training loss (which is not our focus at all). By training loss, we mean the loss of the model on the training set, not the noisy per-iterate estimate of it that drives SGD. We argue that the training loss is very likely to improve monotonically with a not too small period for evaluation. If the training loss does not improve for an entire period, then there is a problem with the underlying optimization (or we reached convergence). Also, we discuss in section 5.1 how the impact of small deviations from monotonicity is limited.
> >
> > You say "We argue that the training loss is very likely to improve monotonically with a not too small period for evaluation" --- what is the argument? Certainly the final iterate of SGD does not have training/validation/test loss which is monotonically improving due to stochasticity. For example, consider minimizing
> >
> > $$f(x) = f_1(x) + f_2(x) = 2x^2 - x^2.$$
> >
> > The minimizer is $x^* = 0$, but the optimizer will move away from $x^*$ and worsen the objective every time $f_2$ is sampled. Thus, both the final iterative and a local average will be non-monotonic in both objective and distance to optimum. Moreover, this problem is easily re-framed as an ERM problem with validation loss that is also not monotone.
> >
> > > Yes, this assumption is of course there, but it is not our invention: the underlying optimization problem (i.e. without averaging) must be set up reasonably to have any hope of generalizing from the training set to validation loss or good downstream performance. More abstractly, this is an assumption of Empirical Risk Minimization (when considering the validation loss). If this assumption does not hold, then the learning problem is badly posed, and how to do averaging is the least of our problems.
> >
> > What about overfitting? Overfitting is a well known issue with ERM and means that improvement on the training loss will not correspond to improvement on the validation loss. I mention this because it's important that the paper explicitly discusses how strong these assumptions are and how they may violate standard behavior in learning problems.
> >
> > > The experimental details are described in a published paper, which we didn't cite to better preserve anonymity. We couldn't provide a complete description either for the same reason. We will update the paper for the camera ready ...
> >
> > Are you saying that the Penn Treebank experiments are reproduced from a previously published paper? If this is the case, then you absolutely must cite this published paper, else you are plagiarizing the results. It does not matter if the plagiarism is self-plagiarism --- it is plagiarism nonetheless. It is also critical that you verify this reproduction is not significant enough to violate the conference's dual submission policy. Finally, you should provide the details necessary to verify/recreate the experiments in this submission, otherwise the submission is not reproducible

---

> > > ### Author Response · Authors · 2022-11-24
> > > **More on Assumption 1 and the experimental setup**
> > >
> > > We attempt to answer the following two (maybe three) comments at the same time.
> > >
> > > > There is a difference between characterizing the function class on which a method works (e.g. Lipschitz continuous functions) and assuming properties which cannot be verified (e.g. monotonicity of the validation loss) on a specific class of functions. It is this latter issue that I am criticizing. I think your assumptions are akin to assuming that gradient descent makes progress on $f$, instead of assuming $f$ is Lipschitz smooth and then deriving a progress condition.
> > >
> > > > What about overfitting? Overfitting is a well known issue with ERM and means that improvement on the training loss will not correspond to improvement on the validation loss. I mention this because it's important that the paper explicitly discusses how strong these assumptions are and how they may violate standard behavior in learning problems.
> > >
> > > Two-tailed averaging relies on the optimization problem and and $f$ being related. It won't fix overfitting if it's present in the non-averaged case (but the reset heuristic will trigger after a while). So this assumption is of course still there implicitly, but rereading this discussion, we feel that there may be a misunderstanding here: these came up while discussing Assumption 1, which does not say that $f$ is monotonically decreasing during optimization, it says that averaging more iterates preceding time step $n$ monotonically improves the averaged weights in terms of $f$.
> > >
> > > > You say "We argue that the training loss is very likely to improve monotonically with a not too small period for evaluation" --- what is the argument? Certainly the final iterate of SGD does not have training/validation/test loss which is monotonically improving due to stochasticity. For example, consider minimizing [snip]
> > >
> > > As this also came from the discussion of Assumption 1, we were talking about the training loss with the averaged weights. If there indeed had been a misunderstanding about Assumption 1 (see above), then does this question still stand?
> > >
> > > > Are you saying that the Penn Treebank experiments are reproduced from a previously published paper? If this is the case, then you absolutely must cite this published paper, else you are plagiarizing the results. It does not matter if the plagiarism is self-plagiarism --- it is plagiarism nonetheless. It is also critical that you verify this reproduction is not significant enough to violate the conference's dual submission policy. Finally, you should provide the details necessary to verify/recreate the experiments in this submission, otherwise the submission is not reproducible
> > >
> > > No, the experiments were not reproduced from a previously published paper. The experimental setup was reused.
> > >
> > > In particular, we used the experimental and hyperparameter tuning setup in "Circling Back to Recurrent Models of Language" (arxiv, 2022) and "Mogrifier LSTM" (ICLR, 2019).
> > > We trained 2-layer Rewired Mogrifier LSTM models with Rectified Adam for about 300 epochs with 2 dropout samples.
> > > Batch size was 256, and we trained with a BPTT window size of 70.
> > > The rest of the hyperparameters were tuned with a black-box gaussian process based tuner as in "Mogrifier LSTM" with the addition of the Chrono init limit for the LSTM forget gate.

---

> > > > ### Comment · Reviewer_NMLB · 2022-11-24
> > > > **Thanks for Clarifying**
> > > >
> > > > 1. Thanks for the clarification around Assumption 1. However, I have not misunderstood what the assumption requires. This is why I said "both the final iterate and a **local average** will be non-monotonic in both objective and distance to optimum". As you expand the length of the average window, the average will worsen and then improve as the "bad" and "good" steps are alternately included in the average. I hope this is clear from the example.
> > > >
> > > > 2. Thanks for providing the experimental details. These should be added to the manuscript.
> > > >
> > > > 3. I would my score to four to reflect our conversation and the improvements made to the manuscript, but I cannot due to the review form. As such, I will maintain my score of three. Although I appreciate this discussion, I am not convinced that the assumptions are reasonable. I believe that more experiments are required to justify the assumptions or an alternative theoretical analysis should be developed before the manuscript is ready for publication.

---

### Official Review · Reviewer_bYZd · 2022-10-27

**Confidence:** 4
**Correctness:** 3
**Technical Novelty And Significance:** 2
**Empirical Novelty And Significance:** Not applicable
**Recommendation:** 3

**Clarity, Quality, Novelty And Reproducibility:**

Clarity is fine.

Quality needs some discussion. Please see my questions above.

Novelty is good.

Reproducibility is good.

**Strength And Weaknesses:**

# Strength
1. Overall I think the paper is written well. I find the problem well motivated, that treating with the hyperparameter in tail-averaging might be painful, so we need a fix to that.
2.  The proposed two-tailed averaging is novel to my knowledge.
3. Both theory and experiments are presented for better demonstrating the advantages of  two-tailed averaging.

# Weakness
1. It is claimed that two-tailed averaging is "adaptive" and "hyperparameter-free". However I find this a bit misleading, because the procedure needs to be able to evaluate the "loss" periodically. Note that the "loss" refers to the targeted objective to be minimized if I understand correctly. For example, in the context of stochastic optimization [Jain et al, 2018], the "loss" refers to the population loss. But being able to evaluate the true loss is a very strong requirement, and makes the point of "hyperparameter-free" less interesting. For example, one can easily come up with a variant of tail-averaging that is "hyperparameter-free" provided the ability to evaluate the loss. Let us consider the setting of [Jain et al, 2018] again, one can simply save all the SGD iterates, and evaluate the loss over all possibilities of tail-averaging (or any general weighted averaging of the iterates), and output the optimal one --- this is "hyperparameter-free" and "optimal". I would like to see some discussions on the requirement of accessing the true loss. I just think this is too strong in many settings.


2. I think the current presentation is somewhat "abstract". Perhaps the authors can discuss a potential application of the proposed two-tailed averaging in some theory/application problem? As the authors mentioned [Jain et al, 2018] many times, could you formally state a bound for two-tailed averaging in the setting of learning least squares with SGD, and compare it to the one obtained by tail-averaging? Being "anytime", "adaptive" and "optimal once in a while" sounds very cool but I would like to see some more specific benefits of using two-tailed averaging.


3. I also think the assumptions could have been checked with a specific exemplar problem. The current assumptions are just very abstract and I do not immediately see how applicable/realistic they are.




**Summary Of The Paper:**

Tail-averaging is a very useful trick, in both theory and practice, to improve the performance of iterative learning/optimization method. Yet, it introduces an important hyperparameter, the starting point for the tail-averaging. This work proposes a two-tailed averaging method as an adaptive and hyperparameter-free alternative to the tail-averaging method. Under suitable conditions, it is proved that two-tailed averaging has many good properties, inclduing being (1) anytime, (2) adaptive, and (3) optimal once in a while. Besides to the method and theory, some experiments are conducted to verify the advantage of two-tailed averaging.

**Summary Of The Review:**

Please see above. Due to the mentioned issues I would like to keep my score low. I would like to hear the authors rebuttal to see whether or not I have misunderstood anything.

---

> ### Author Response · Authors · 2022-11-18
> **Response to bYZd**
>
> Thank you for the well-considered review.
> We incorporated much of the feedback to improve the paper.
> See our responses to specific questions below, but before that we would like to address a misunderstanding of two-tailed averaging that's common to the reviews.
>
> **Focus on Generalization**: In the original submission, we did not make it clear at all that our primary concern is generalization.
> Both the title and the abstract suggested a focus on stochastic optimization, which was further reinforced by the "Related Works" section.
> Our concern is with the minimization of some generalization loss $f$ on top of stochastic optimization of a training loss.
> The two losses are of course related (and must be for this kind of training to work even without averaging), but our method is fairly agnostic to optimization details.
> The goal of two-tailed averaging is to improve generalization (i.e. minimize $f$), not to minimize the training loss.
> We changed the title to reflect this from "... Iterate Averaging for Stochastic Optimization" to "... Weight Averaging for Better Generalization".
> We also updated the abstract, the Related Works section, and other parts of the paper to fix this.
>
> > being able to evaluate the true loss is a very strong requirement
>
> We don't assume access to the true loss only to a generalization loss (e.g. validation loss or downstream task performance), which is overwhelmingly common outside pure optimization settings.
> Same with periodic evaluation: we are taking advantage of what is fairly common practice.
> We updated the paper to reflect this.
>
> > hyperparameter-free is less interesting with periodic eval (there is a trivial algorithm)
>
> This is a very important point, and we added discussion of this to Section 3 "Problem Statement".
> The trivial algorithm of saving all iterates and searching for the optimal length has a storage and evaluation cost proportional to current time step t.
> If we make Assumption 1, then we can stop once the things start to worsen, and the cost will be proportional to the length of the optimal solution.
> Crucially, TTA approximates this with a constant cost.
>
> > could you formally state a bound for two-tailed averaging in the setting of learning least squares with SGD, and compare it to the one obtained by tail-averaging?
>
> We agree that this would be a very nice thing to have from an optimization perspective, but given our focus on generalization, we did not pursue this.
>
> > I also think the assumptions could have been checked with a specific exemplar problem. The current assumptions are just very abstract and I do not immediately see how applicable/realistic they are.
>
> We relaxed Assumption 1 (the one about monotonicity in averaging length) a bit to better reflect what the proofs need.
> We also cleared up the presentation of the reset heuristic.
> With that, in the updated paper, there are 4 assumptions, each with an intuitive meaning.
> 1. Averaging over more weights monotonically improves the loss until the optimum.
> 2. The loss right after optimal length (round up to evaluation period) is better than anything longer.
> 3. The optimal averaging length grows sublinearly during training.
> 4. The optimal averaging length monotonically increases during training.
> We also updated the section "On Failed Assumptions" a bit.
>
> All in all, we feel that while abstract, these assumptions make intuitive sense given our current (sketchy!) understanding of stochastic gradient descent especially in the context of sizable neural networks.

---

### Decision · Program_Chairs · 2023-01-20

**Decision:**

Reject

**Justification For Why Not Higher Score:**

The unverified assumption makes the entire theory not very useful.

**Justification For Why Not Lower Score:**

N/A

**Metareview: Summary, Strengths And Weaknesses:**

The paper seems to rely on an unverified assumption about the behavior of the value of a stochastic function evaluated on averaged iterates of SGD. The assumption seems to be critical for the analysis and indeed it seems to have been proposed exactly to make the analysis go through. There is no empirical evaluation to verify when this assumption holds nor a theoretical argument to show benign conditions that imply this assumption. These issues have not been solved despite a detailed discussion with the reviewers.
Hence, the paper might have potential but it seems to be built on shaky grounds and as such it cannot be accepted at ICLR.